# Proton-selective coating enables fast-kinetics high-mass-loading cathodes for sustainable zinc batteries

Quanquan Guo[1,2,3,12], Wei Li [1,4,12], Xiaodong Li [1,2,12], Jiaxu Zhang[1], Davood Sabaghi[1], Jianjun Zhang[1], Bowen Zhang[5], Dongqi Li [1], Jingwei Du[1], Xingyuan Chu[1], Sein Chung [6], Kilwon Cho [6], Nguyen Ngan Nguyen[1,2], Zhongquan Liao[5], Zhen Zhang [7], Xinxing Zhang [3], Grégory F. Schneider [8], Thomas Heine [9,10,11], Minghao Yu [1]✉ & Xinliang Feng [1,2]✉

The pressing demand for sustainable energy storage solutions has spurred the burgeoning development of aqueous zinc batteries. However, kinetics-sluggish $Zn^{2+}$ as the dominant charge carriers in cathodes leads to suboptimal charge-storage capacity and durability of aqueous zinc batteries. Here, we discover that an ultrathin two-dimensional polyimine membrane, featured by dual ion-transport nanochannels and rich proton-conduction groups, facilitates rapid and selective proton passing. Subsequently, a distinctive electrochemistry transition shifting from sluggish $Zn^{2+}$-dominated to fast-kinetics $H^+$-dominated Faradic reactions is achieved for high-mass-loading cathodes by using the polyimine membrane as an interfacial coating. Notably, the $NaV_3O_8 \cdot 1.5H_2O$ cathode (10 mg cm$^{-2}$) with this interfacial coating exhibits an ultrahigh areal capacity of 4.5 mAh cm$^{-2}$ and a state-of-the-art energy density of 33.8 Wh m$^{-2}$, along with apparently enhanced cycling stability. Additionally, we showcase the applicability of the interfacial proton-selective coating to different cathodes and aqueous electrolytes, validating its universality for developing reliable aqueous batteries.

Rechargeable aqueous batteries have emerged as an attractive sustainable technology for grid-scale energy storage because of their advantages in safety, cost efficiency, scalability, and low environmental impacts[1]. Of particular interest are aqueous zinc batteries (AZBs), which directly take cheap zinc metal as the capacity-dense (820 mAh g$^{-1}$ and 5855 mAh cm$^{-3}$) and low-redox-potential (−0.76 V vs. standard hydrogen electrode) anodes, and mild acidic aqueous Zn solutions as the highly ionic conductive (up to 1 S cm$^{-1}$), nonflammable, and nontoxic electrolytes[2,3]. Intensive efforts have been dedicated to overcoming the critical barriers associated with the Zn stripping/plating anode chemistry[4–8], including dendrite formation, corrosion, and hydrogen evolution. It is encouraging to see that considerable

[1]Center for Advancing Electronics Dresden (cfaed) & Faculty of Chemistry and Food Chemistry, Technische Universität Dresden, Dresden, Germany. [2]Max Planck Institute of Microstructure Physics, Halle (Saale), Germany. [3]State Key Laboratory of Polymer Materials Engineering, Polymer Research Institute, Sichuan University, Chengdu, China. [4]State Key Laboratory of Applied Organic Chemistry, College of Chemistry and Chemical Engineering, Lanzhou University, Lanzhou, PR China. [5]Fraunhofer Institute for Ceramic Technologies and System (IKTS), Maria-Reiche-Straße 2, Dresden, Germany. [6]Department of Chemical Engineering, Pohang University of Science and Technology, Pohang, South Korea. [7]School of Chemistry and Materials Science, University of Science and Technology of China, Hefei, China. [8]Leiden Institute of Chemistry, Leiden University, P.O. Box 9502 Leiden, The Netherlands. [9]Theoretical Chemistry, Technische Universität Dresden, Dresden, Germany. [10]Institute of Resource Ecology, Helmholtz-Zentrum Dresden-Rossendorf, Leipzig Research Branch, Leipzig, Germany. [11]Department of Chemistry, Yonsei University, Seodaemun-gu Seoul, Korea. [12]These authors contributed equally: Quanquan Guo, Wei Li, Xiaodong Li. ✉e-mail: minghao.yu@tu-dresden.de; xinliang.feng@tu-dresden.de

progress has been made in recent studies on the development of current collectors[9], interphases[10,11], and electrolytes[12–15], which have enabled practical implementation of Zn anode chemistry with promising feasibility. On the other hand, high-performance cathodes are indispensable for assembling advanced AZB devices. Various cathode materials capable of accommodating $Zn^{2+}$, with metal oxides (e.g., V-, Mn-, and Mo-based oxide compounds) as high-voltage and high-capacity materials, have been explored to couple with the Zn metal anode[16–18]. Nevertheless, these cathode chemistries are often governed by the dominant $Zn^{2+}$ insertion/extraction and partial $H^+$ insertion/extraction[19]. The bivalent nature of $Zn^{2+}$ introduces significant barriers associated with interfacial desolvation (from $Zn(H_2O)_6^{2+}$), solid-state diffusion, and hosting density, which significantly restricts the charge-storage kinetics of cathodes[20], especially at practical mass loading (>5 mg cm$^{-2}$)[21–23]. As a result, high-loading cathodes frequently fail to fully express their theoretical promise, presenting suboptimal capacity, rate capability, and durability.

The choice of charge carrier ions has a significant impact on the Faradaic reaction kinetics of battery electrodes[24]. In AZBs, $H^+$ or its hydrated form ($H_3O^+$) offers more favourable kinetics compared to $Zn^{2+}$ owing to the smaller size (1.2 Å for $H^+$, 2.8 Å for $H_3O^+$), lighter mass (1.0 g mol$^{-1}$ for $H^+$, 19.0 g mol$^{-1}$ for $H_3O^+$), as well as rapid conduction in aqueous electrolytes governed by the Grotthuss mechanism[25]. However, increasing the involvement of $H^+$ in the cathode chemistries is challenging, as the $H^+$ concentration in the mild acidic electrolyte of AZBs is typically several orders of magnitude lower than the $Zn^{2+}$ concentration[26]. One approach to enhance $H^+$ participation is to incorporate hydrogen-bonding networks into electrode structures, which allows for proton Grotthuss conduction to occur within solid electrodes and promotes $H^+$-selective insertion/extraction[27,28]. Recent studies have demonstrated the effectiveness of this design strategy in layered cathode materials by confining species like $H_2O$[29] and $NH_4^+$[30] in the interlayer space. Additionally, crystal structural regulation through methods such as introducing vacancies[31], doping[32], or pre-intercalation of different guest ions[33] has shown promise in enhancing the $H^+$ contribution to the overall charge storage in certain cathode structures, although the underlying driving forces remain ambiguous. While important progress has been made, these material modification approaches are often limited to specific cathode structures and have primarily been validated for electrodes with low mass loadings (<2 mg cm$^{-2}$). It remains a significant challenge to develop a universal strategy for controlling charge carriers of cathodes in extensive AZB systems, overcoming limitations related to the type of electrodes and electrolytes.

In this study, we achieve the successful electrochemistry transition for high-mass-loading AZB cathodes from sluggish $Zn^{2+}$-dominated to fast-kinetics $H^+$-dominated Faradic reactions by using a proton-selective coating strategy (the $H^+/Zn^{2+}$ ratio increases from 0.4 to 3.5). We discover that two-dimensional polyimine membrane (2DPM) with well-defined dual ion-transport channels and plentiful $H^+$-conductive sites can enable the universal interfacial proton-selective coating. The high-flux imine-enclosed nanochannel and proton-selective porphyrin-based nanochannel design endows 2DPM with a high $H^+$ flux exceeding 0.9 mol m$^{-2}$ h$^{-1}$ and impressive $H^+$ transport selectivity of 140.7 over $Zn^{2+}$ at a thickness of 80 nm. Such a favourable ion transport feature of 2DPM enables the interfacial coating exhibiting preferential transport of $H^+$ and maximising the passing discrimination of $H^+$ and $Zn^{2+}$ at the cathode/electrolyte interface (Fig. 1a). When coupled with a high-mass-loading $NaV_3O_8 \cdot 1.5H_2O$ electrode (denoted NVO, 10 mg cm$^{-2}$), 2DPM empowers the electrode with significantly improved specific capacity (from 288.8 to 450.5 mAh g$^{-1}$; theoretical capacity is 485.6 mAh g$^{-1}$), exceptionally high areal capacity of 4.5 mAh cm$^{-2}$ and state-of-the-art energy density of 33.8 Wh m$^{-2}$. 2DPM is also shown to alleviate the structural distortion of NVO, thus contributing to enhanced cycling stability (68.6% vs. 87.8% after 1000

cycles) compared to pristine NVO electrode. We further demonstrate the universality of proton-selective 2DPM coating in boosting the charge-storage kinetics of other cathodes (e.g., $\varepsilon$-$MnO_2$ and $\alpha$-$MoO_3$) in different electrolytes (e.g., 2 M $ZnSO_4$ and 20 m $ZnCl_2$).

## Results

### $H^+$ and $Zn^{2+}$ transport properties of 2DPM

The ideal interfacial coating for AZB cathodes should meet several criteria: (1) maintain structural stability during the harsh electrochemical process; (2) facilitate a significant ion transport flux; and (3) allow for selective $H^+$ passing. With these considerations in mind, layer-stacked 2D crystalline polymers are considered promising candidates owing to their desirable features, such as robust structural stability, well-defined nanochannels, and tailorable scaffold functionalities. In this study, we designed a crystalline 2DPM membrane with dual ion-transport nanochannels (Fig. 1b), which can be achieved by the Schiff-base polycondensation of 5, 10, 15, 20-tetrakis (4-aminophenyl) porphyrin (TAPP) and 2,5-Dihydroxyterephthalaldehyde (DHTAP). Specifically, 2DPM possesses both imine-enclosed nanochannels (2.5 nm in size) and porphyrin centre nanochannels (around 3.4 Å in size), and the areal density of ion-transport nanochannels reaches a high level of $10^{17}$ m$^{-2}$ (Supplementary Fig. 1). Moreover, the incorporated hydroxyl side groups, imine linkages, and porphyrin pyrrole units on the nanochannel wall of 2DPM all serve as desirable proton-conductive sites with a high density of about $3 \times 10^{27}$ m$^{-3}$ (Supplementary Fig. 2)[24,29,30].

Large-area thin film of 2DPM (up to 28 cm$^2$, Fig. 1c) with tunable thickness (20–100 nm, Supplementary Fig. 3) was synthesized through a scalable surfactant-monolayer-assisted interfacial synthesis route (Supplementary Fig. 4)[34]. Its high crystallinity and face-on orientation were confirmed by the high-resolution transmission electron microscopy (HR−TEM) images and the corresponding selected-area electron diffraction (SAED) pattern (Fig. 1d, Supplementary Fig. 5). Grazing incidence wide-angle X-ray scattering (GIWAXS) measurement of 2DPM detected sharp and discrete Bragg peaks near the $Q_Z = 0$ position (Fig. 1e), which supports high in-plane crystallinity. All the GIWAXS peaks match perfectly with an inclined AA-stacking model simulated using London dispersion-corrected density functional theory (DFT) calculations (Supplementary Fig. 6). Besides, the simulated electronic band structure identifies a giant band gap for 2DPM (1.53 eV), indicating its poor electron-conductive nature (Supplementary Fig. 7).

We started with the thickness optimization for 2DPM to reach the balanced $H^+$ conduction and permselectivity. The membranes with thicknesses of 20, 60, 80, and 100 nm are denoted 2DPM-20, 2DPM-60, 2DPM-80, and 2DPM-100, respectively. We first evaluated the $H^+$-transmembrane transport behaviour of 2DPM by the concentration-driven permeation measurement with a setup[35] illustrated in Supplementary Fig. 8. The commercial nylon microporous membrane with ultra-large ion permeation flux and no ion-transport selectivity was applied to support 2DPM (Supplementary Fig. 9). $Zn^{2+}$-transmembrane transport, as the competition with $H^+$ transport in AZBs, was also assessed for 2DPM with the same setup. Fig. 2a, b plot the $H^+$ and $Zn^{2+}$ permeation curves of 2DPM as a function of time, respectively. All the membranes follow the linear permeation relationship with constant transport rates for both $H^+$ and $Zn^{2+}$. It is notable that 2DPM in the thickness range of 20–80 nm depicts almost thickness-independent $H^+$ transport, showing a high $H^+$ permeation rate of 0.91–0.95 mol m$^{-2}$ h$^{-1}$ and an excellent proton diffusion coefficient of $6.2 \times 10^{-7}$–$6.5 \times 10^{-7}$ cm$^2$ s$^{-1}$. When the membrane thickness reaches 100 nm, the $H^+$ permeation rate slightly decreases to 0.73 mol m$^{-2}$ h$^{-1}$. By contrast, $Zn^{2+}$ transport through 2DPM heavily depends on the membrane thickness, and the $Zn^{2+}$ permeation rates for 2DPM-20, 2DPM-60, 2DPM-80, and 2DPM-100 are 0.51, 0.27, 0.0065 and 0.0062 mol m$^{-2}$ h$^{-1}$, respectively. Fig. 2c further summarizes the ion permeation rates of all the membranes and defines the $H^+/Zn^{2+}$ permeation rate ratio as the ion-transport selectivity. As revealed, 2DPM-80 exhibits the best $H^+/Zn^{2+}$ selectivity of

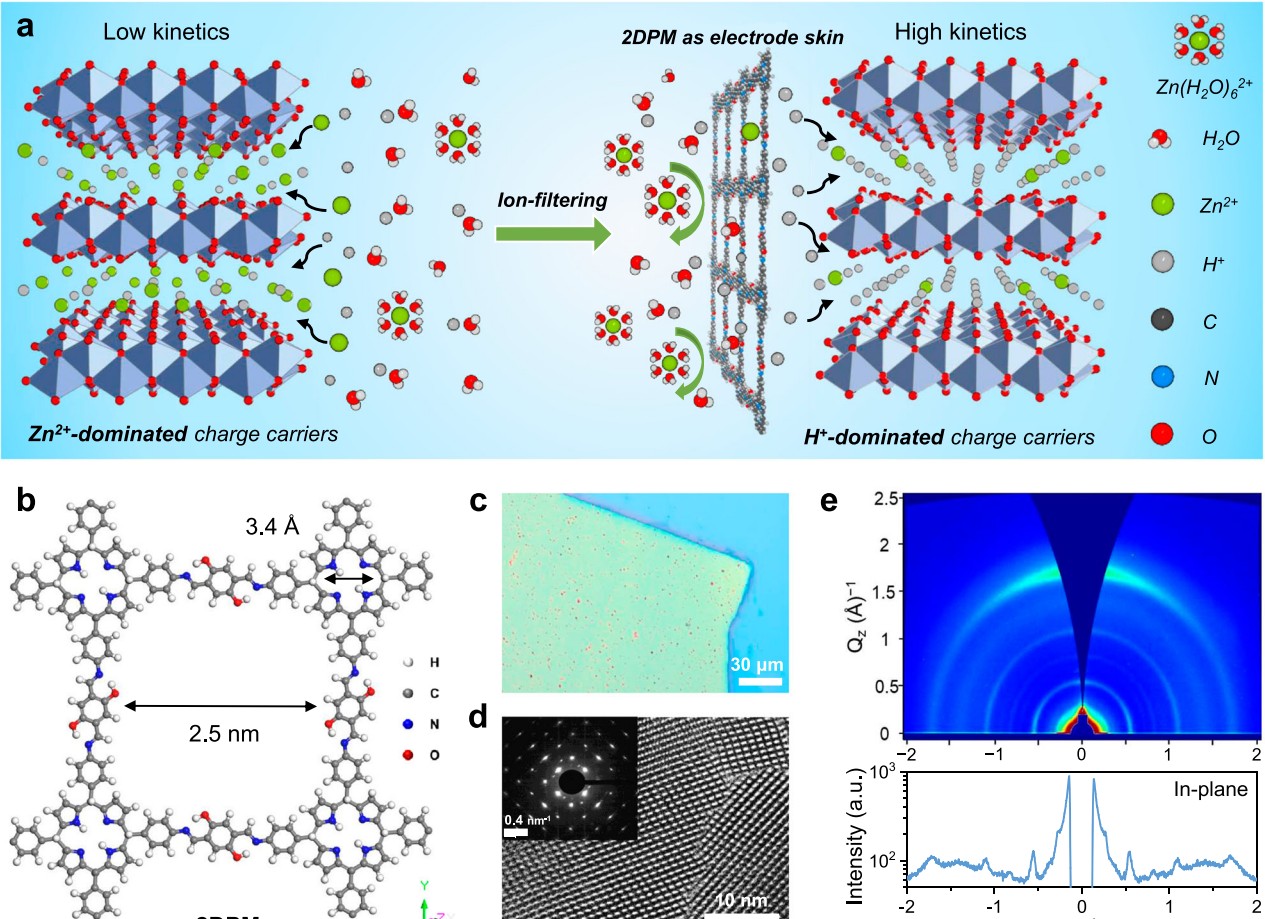

**Fig. 1 | Structural characterisations of 2DPM. a** Schematic illustration showing the H⁺-dominated cathode intercalation chemistry enabled by the H⁺-selective 2DPM coating. **b** The simulated structure of 2DPM with dual-ion transport nanochannels, including imine-enclosed nanochannels (2.5 nm in size) and porphyrin centre nanochannels (around 3.4 Å in size). **c** Optical microscopy image. **d** High-resolution transmission electron microscopy image. The inset shows the corresponding selected-area electron diffraction pattern. **e** Grazing incidence wide-angle X-ray scattering and the corresponding in-plane pattern of 2DPM with a thickness of 80 nm.

140.7, while maintaining a high H⁺ permeation rate. Moreover, the areal ion conductivity of 2DPM-80 in 0.5 M $H_2SO_4$ was measured to be 45.8 mS cm⁻² (Supplementary Fig. 10), which is higher than monolayer graphene (12 mS cm⁻²) and bilayer boron nitride (40 mS cm⁻²)[36].

In addition, the H⁺/Zn²⁺ selectivity of 2DPM-80 was evaluated by filling the seed chamber with a conventional ZAB electrolyte (i.e., 2 M $ZnSO_4$, pH = 4.3). The H⁺ concentration and Zn²⁺ concentration in the permeated chamber were simultaneously detected at regular intervals over time (Fig. 2d). It is worth noting that the H⁺ concentration in the mild acidic $ZnSO_4$ electrolyte is only $5 \times 10^{-5}$ M, five orders of magnitudes lower than its Zn²⁺ concentration (2 M). Despite this, it is notable that 2DPM-80 still depicts a considerably superior H⁺ permeation rate (0.046 mol m⁻² h⁻¹) to the Zn²⁺ permeation rate (0.013 mol m⁻² h⁻¹), providing an excellent H⁺/Zn²⁺ selectivity of 3.5. All these results suggest that 2DPM is preferential for transporting proton and can effectively block most Zn²⁺ diffusion.

To understand the effect of two different types of nanochannels of 2DPM on ion transport properties, we further synthesized a control 2DP membrane with a thickness of 60 nm (denoted Cu-2DPM-60) by using 5, 10, 15, 20-tetrakis (4-aminophenyl) porphyrin-Cu (II) instead of the metal-free TAPP monomer (Supplementary Fig. 11). In comparison with 2DPM-60, Cu-2DPM-60 presents a sharply declined H⁺ permeation rate (0.58 vs. 0.92 mol m⁻² h⁻¹), but maintains a comparable one for Zn²⁺ (0.28 vs. 0.27 mol m⁻² h⁻¹). This ion-transport difference mainly stems from the chemical nature of Cu-2DPM, in which the

central nanochannels of the porphyrin pore are blocked. This result thus reflects that the metal-free porphyrin units significantly contribute to the selective H⁺ conduction, while almost completely blocking the Zn²⁺ transport. DFT simulations are conducted to offer insights into this distinct capability of 2DPM in transport of H⁺ and Zn²⁺. The electrostatic potential (ESP) plots of 2DPM reveal that certain regions rich in electrons, such as the O atoms of phenolic hydroxyl groups, N atoms of imine bonds, porphyrin pyrrole units, serve as favourable sites for cation hopping (Fig. 2e). Specifically, Fig. 2f illustrates the H⁺/Zn²⁺ transport paths along the imine-enclosed nanopores, as well as the interactions between H⁺/Zn²⁺ with the adjacent functional groups at the initial, transition, and final transport stages. The calculated energy profiles (Fig. 2g) demonstrate that phenolic hydroxyl and imine groups jointly assist in the rapid transport of H⁺ with a much lower energy barrier (0.80 eV) than the Zn²⁺ transport (1.29 eV). Additionally, the porphyrin centre excels in H⁺ transport with an extremely low energy barrier of 0.13 eV (Supplementary Fig. 12), while Zn²⁺ shows a high transport energy barrier of 1.34 eV, indicating that Zn²⁺ can hardly pass through the porphyrin pore (Supplementary Fig. 13). This finding is consistent with the experimental observation of contrast transport properties between 2DPM-60 and Cu-2DPM-60.

## Regulation of charge carriers for AZB cathodes
Encouraged by the excellent selectivity toward H⁺ transport of 2DPM, we evaluated 2DPM as the interfacial coating for NVO as the standard

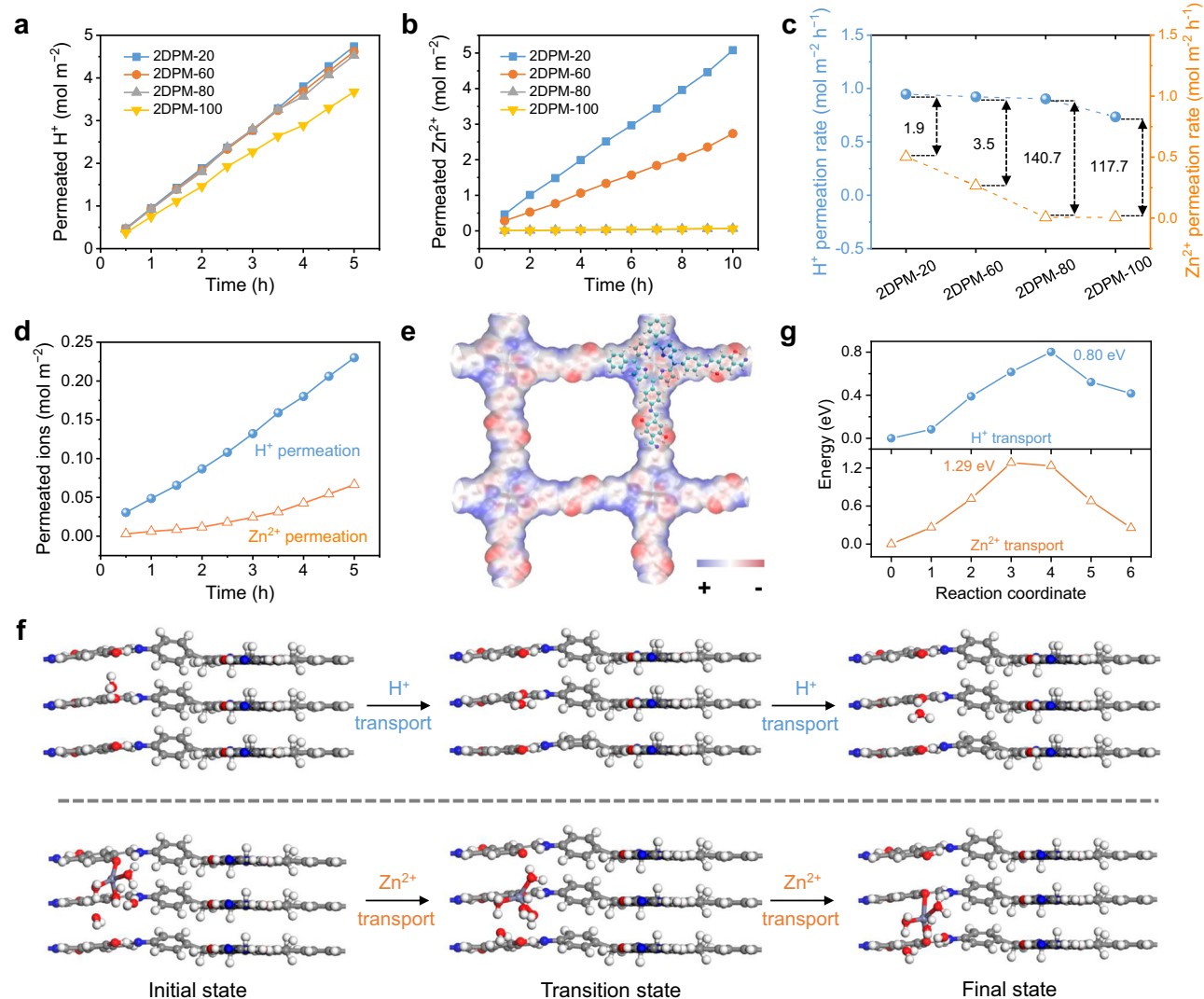

**Fig. 2 | Ion transport properties. a** H+ and **b** Zn2+ permeation curves in the concentration-driven permeation measurements of 2DPM-20, 2DPM-60, 2DPM-80, and 2DPM-100. **c** Ion permeation rates and H+/Zn2+ selectivity of various 2DPM membranes. **d** H+ and Zn2+ permeation curves of 2DPM-80 in 2 M ZnSO4. **e** The electrostatic potential plots of 2DPM. The red and blue colours represent electron-rich and electron-deficient regions, respectively (isosurface value = 0.01 e.Å−3). **f** The simulated H+/Zn2+ transport path and **g** the corresponding energy profiles through the imine-enclosed nanochannels of 2DPM (White: H, Grey: C, Red: O, Blue: N).

AZB cathode, envisioning its advantageous role in directing rapid and selective H+ transport. High-mass-loading NVO (10 mg cm−2) electrodes were prepared based on hydrated sodium vanadate (NaV3O8·1.5H2O, JCPDS no. 16-0601) nanowires (Supplementary Fig. 14). The 1D nano-sized morphology of NVO is expected to shorten the ion diffusion length during charge storage. We transferred the 2DPM membranes onto the NVO surface. Taking 2DPM-80 for instance, the membrane with a negligible mass of 0.04 mg cm−2 allows for the tight and con-formal coating on the NVO surface (Fig. 3a and Supplementary Fig. 15). All electrodes were assessed in two-electrode Swagelok cells with Zn metal as the counter electrode and 2 M ZnSO4 as the electrolyte (Supplementary Fig. 16).

The constructed cathodes with the 2DPM coating were quickly screened by the galvanostatic charge-discharge (GCD) measurements at various current densities (Supplementary Figs. 17 and 18). The cal-culated specific capacities of all electrodes are summarized in (Fig. 3b). All the 2DPM membranes could boost the charge-storage capability of NVO, and the improvement degree of the specific capacity follows the trend of 2DPM-20 < 2DPM-60 < 2DPM-100 < 2DPM-80. This trend is consistent with the H+/Zn2+ selectivity trend illustrated in Fig. 2c. To

confirm the charge carriers in different electrodes, we quantified the Zn/V atomic ratio of all the electrodes at the fully discharged stage (i.e., 0.3 V vs. Zn/Zn2+) by inductively coupled plasma atomic emission spectroscopy (ICP-AES, Supplementary Fig. 19). Meanwhile, the quan-tity of H+ charge carriers is estimated by considering the Zn/V atomic ratio and the total charge transfer per V atom. Fig. 3c displays the charge carrier ratios (H+/Zn2+) of all the electrodes. The contribution of H+ charge carriers to the total charge storage of different electrodes matches well with the H+/Zn2+ selectivity trend of the employed 2DPM. Specifically, 2DPM-80 empowers VNO with the largest H+/Zn2+ ratio of 3.5, which contrasts with the pristine VNO with a low H+/Zn2+ ratio of 0.4. The high charge carrier ratio of H+/Zn2+exactly equal to the H+/Zn2+ selectivity of 2DPM-80 in 2 M ZnSO4 as measured in Fig. 2d. Moreover, we evaluated the electrolyte pH evolution during a discharge/charge cycle of NVO covered by 2DPM-80. As expected, the insertion of H+ into the cathode causes a slight pH increase within a range of 4−6 (Supplementary Fig. 20).

We then seek to gain a comprehensive understanding of the charge carrier species by employing the electrochemical quartz crystal microbalance (EQCM) technique[37]. In this measurement, an NVO-

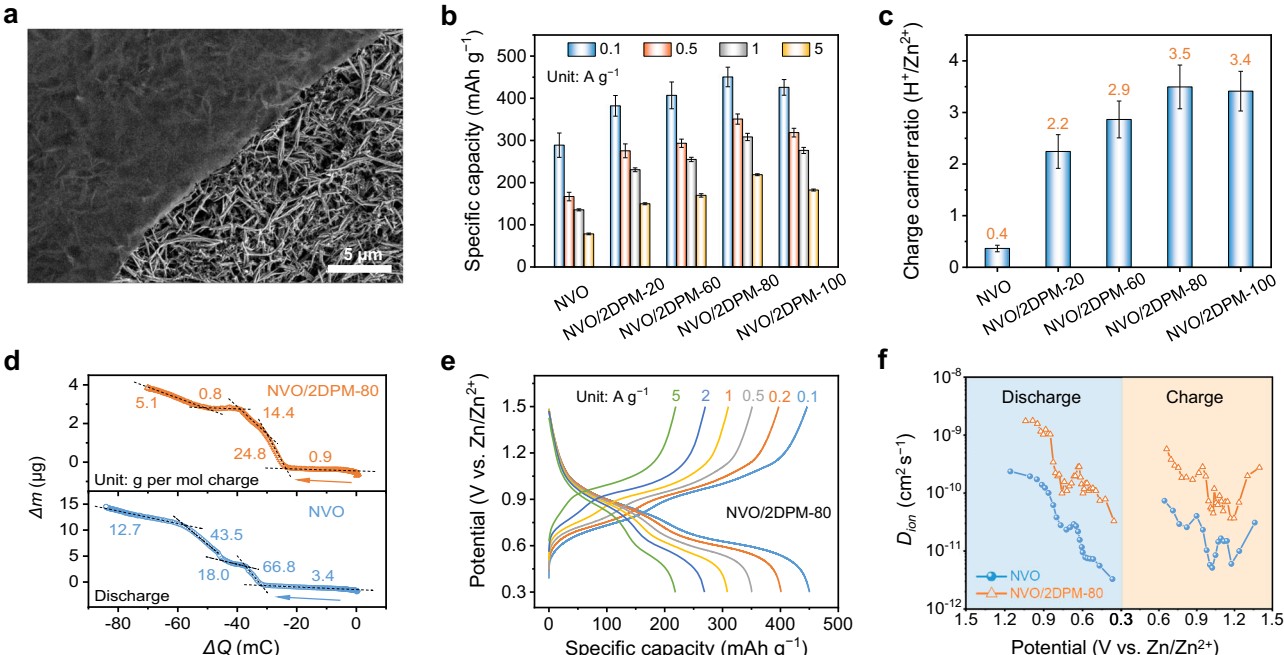

**Fig. 3 | Effects of 2DPM as ion-filtering coating of NVO. a** Scanning electron microscopy image of NVO covered by 2DPM-80. **b** Specific capacities at various current densities and **c** the charge carrier ratios ($H^+/Zn^{2+}$) of NVO covered by 2DPM with different thicknesses. The error bars represent the standard deviation calculated from five in-parallel tests. **d** $\Delta m$ as a function of $\Delta Q$ for NVO and 2DPM-80-

covered NVO in the electrochemical quartz crystal microbalance measurement during discharge. **e** Galvanostatic charge-discharge curves of NVO/2DPM at different current densities. **f** The ion diffusion coefficient ($D_{ion}$) derived from the galvanostatic intermittent titration technique measurement at different potentials of NVO and NVO/2DPM.

coated resonator served as the working electrode in a two-electrode EQCM cell with a Zn foil counter electrode and a 2 M $ZnSO_4$ electrolyte. Cyclic voltammetry (CV) measurements at $1\,mV\,s^{-1}$ were conducted to probe the dynamic mass evolution during the charge/discharge process (Supplementary Fig. 21a, b). The mass change ($\Delta m$) as a function of the charge change ($\Delta Q$) is plotted for both NVO and NVO covered with 2DPM-80 (Fig. 3d, Supplementary Fig. 21c). During ion insertion (i.e., discharge process), five distinct steps were detected for NVO according to molar weights of charge carrier species (i.e., 3.4, 66.8, 18.0, 43.5, and 12.7 g per mol charge). It is notable that ion extraction does not follow a reversible process of ion insertion, showing only three main steps with three types of charge carrier species (i.e., 27.0, 14.8, and 50.8 g per mol charge). With 2DPM-80 coating, NVO exhibited similar five-step ion insertion (i.e., 0.9, 24.8, 14.4, 0.8, and 5.1 g per mol charge) and three-step ion extraction (i.e., 5.4, 2.4, and 32.8 per mol charge). The molar weight of charge carrier species at each insertion or extraction step considerably decreased compared with pristine NVO. More importantly, non-hydrated $H^+$ insertion can solely be recognized in the case of 2DPM-80-covered NVO. It is evidenced by the observation of two ion insertion steps, wherein the charge carrier species possess a molar weight close to 1 g per mol charge. These findings indicate that 2DPM as the interfacial coating not only promotes the $H^+$ as the dominant charge carriers but also mitigates the co-insertion of $H_2O$ into NVO.

### Boosted charge-storage kinetics

To highlight the performance benefits brought by 2DPM, we compared NVO with 2DPM-80-covered NVO (denoted NVO/2DPM). GCD curves indicate that both electrodes present similar charge/discharge plateaus (Fig. 3e and Supplementary Fig. 18), corresponding to the $V^{5+}/V^{4+}$ and $V^{4+}/V^{3+}$ Faradic redox reactions[38]. Despite a high mass loading of $10\,mg\,cm^{-2}$, NVO/2DPM achieves a high specific capacity of $450.5\,mAh\,g^{-1}$, which represents a 56% capacity enhancement compared with NVO ($288.8\,mAh\,g^{-1}$) and approaches

to the theoretical limit of NVO ($485.6\,mAh\,g^{-1}$) considering the $V^{5+}/V^{3+}$ redox reaction. The enhanced capacity is not due to the charge storage capability of 2DPM-80, as shown by the CV comparison of pristine 2DPM-80, NVO, and NVO/2DPM (Supplementary Fig. 22). Moreover, 2DPM/NVO delivers exceptionally high areal capacity of $4.5\,mAh\,cm^{-2}$ and areal energy density of $33.8\,Wh\,m^{-2}$, which significantly outclass the recently reported AZB cathodes, such as V-, Mn-, Mo-, $Fe(CN)_6^{3+}$-, and Prussian blue-based compounds (~$2.1\,mAh\,cm^{-2}$ and ~$18.2\,Wh\,m^{-2}$, Supplementary Table 1). The areal metrics of 2DPM/NVO could be further boosted by enlarging the mass loading of NVO (Supplementary Fig. 23). Moreover, NVO/2DPM demonstrates outstanding rate performance. Even with a 50-fold increase in current density from 0.1 to $5\,A\,g^{-1}$, it maintains a high specific capacity of $228.2\,mAh\,g^{-1}$ ($2.3\,mAh\,cm^{-2}$), referring to the high capacity retention of 50.7%. We also carried out the kinetics analyses of NVO and NVO/2DPM by collecting their CV profiles at a variety of scan rates (Supplementary Figs. 24 and 25), which all support the boosted charge-storage kinetics of NVO/2DPM.

To understand the origin of the high-kinetics performance of NVO/2DPM, the diffusivity of charge carrier ions in NVO and NVO/2DPM was evaluated using the galvanostatic intermittent titration technique (GITT, Supplementary Fig. 26). Fig. 3f compares the ion diffusion coefficient ($D_{ion}$) of both electrodes during the charge and discharge process. Of note, $D_{ion}$ of NVO/2DPM ($3.1 \times 10^{-11}$–$1.7 \times 10^{-9}\,cm^2\,s^{-1}$) is approximately one order of magnitude higher than that of NVO ($3.1 \times 10^{-12}$ ~ $2.2 \times 10^{-10}\,cm^2\,s^{-1}$) at similar charge/discharge states. This result indicates that the increased involvement of $H^+$ promotes solid-state ion diffusion in the electrode. Moreover, the obviously reduced equivalent series resistance (ESR) and charge-transfer resistance ($R_{ct}$) in NVO/2DPM was identified by the electrochemical impedance spectroscopy (EIS) measurements of both electrodes at various potentials (Supplementary Fig. 27), manifesting the decreased internal resistance and improved charge-transfer efficiency of NVO/2DPM associated with the enriched $H^+$ as charge carriers.

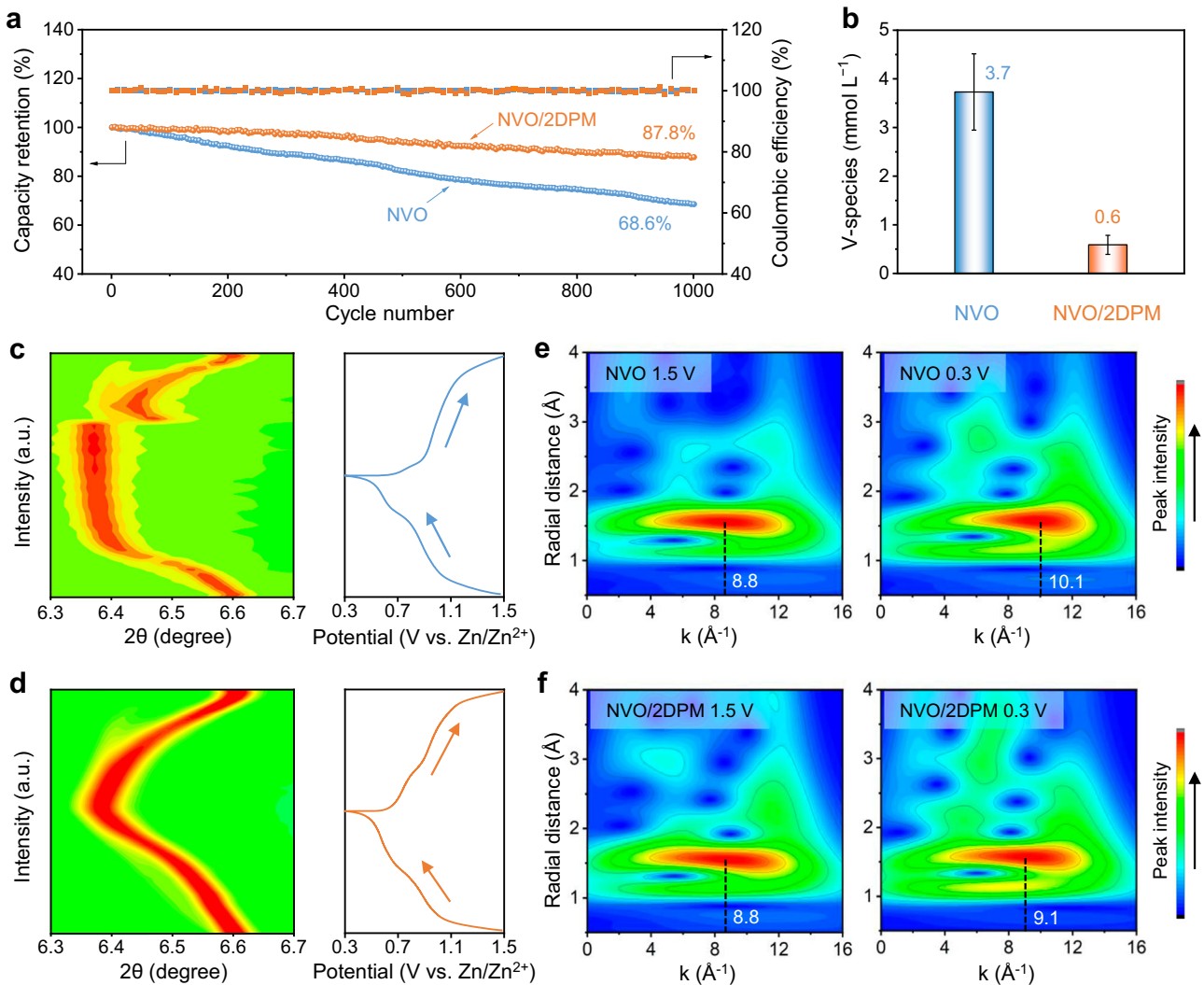

**Fig. 4 | Cycling performance and structure evolution. a** Cycling performance of NVO (Blue line) and NVO/2DPM (Yellow line) at 3 A g$^{-1}$. **b** The dissolved V-species concentration in the electrolyte after 10 charge/discharge cycles at 0.1 A g$^{-1}$. Operando synchrotron X-ray diffraction measurement of **c** NVO and **d** NVO/2DPM during one GCD cycle. The contour plots of wavelet-transformed extended X-ray absorption fine structure spectra of **e** NVO and **f** NVO/2DPM at fully charged (1.5 V vs. Zn/Zn$^{2+}$) and discharged (0.3 V vs. Zn/Zn$^{2+}$) states.

## Promoted cycling stability

Apart from the enhanced charge-storage kinetics, we further evaluated the cycling stability of NVO/2DPM. In a long-term cycling test at 3 A g$^{-1}$, NVO/2DPM sustains 87.8% of its original capacity after 1000 charge/discharge cycles with nearly 100% coulombic efficiencies (Fig. 4a). By contrast, the capacity of NVO fast decays to 68.6% of the original value. Even at a low current density of 1 A g$^{-1}$, NVO/2DPM shows enhanced cycling stability compared with NVO (Supplementary Fig. 28). After the cycling test, NVO/2DPM was disassembled from the cell and subjected to scanning electron microscopy (SEM) and Fourier transform infrared spectroscopy (FTIR) characterisations (Supplementary Fig. 29). As revealed, 2DPM remains tightly covering the NVO surface with all characteristic FTIR peaks detected, verifying the robust chemical stability of 2DPM during repeated charge/discharge cycles. The excellent electrochemical stability of 2DPM was also confirmed by the GCD measurement of individual 2DPM (Supplementary Fig. 30). Additionally, using the ICP-AES measurements, we measured the concentration of V-species in the electrolyte after 10 charge/discharge cycles of NVO and NVO/2DPM at 0.1 A g$^{-1}$. The concentration of 0.6 mmol L$^{-1}$ for NVO/2DPM is substantially lower than the 3.7 mmol L$^{-1}$ for NVO alone (Fig. 4b). This result suggests that the 2DPM coating can protect the

integrity of NVO by inhibiting the loss of active material due to intermediate dissolution. As a simple proof-of-concept, we also demonstrated a pouch-cell AZB device (2.5 × 4 cm$^2$) with a two-layer NVO/2DPM cathode and a Zn foil anode (Supplementary Fig. 31). The device achieved a capacity of 65.8 mAh at 0.5 A g$^{-1}$ and can be operated stably for 280 cycles at 1 A g$^{-1}$.

We then used the operando synchrotron X-ray diffraction measurement (XRD, wavelength: 0.20733 Å) to understand the enhanced cycling performance of NVO/2DPM (Supplementary Fig. 32). We particularly focused on the shift of the characteristic (204) peak, which is associated with the spacing change between V$_3$O$_8$ layers due to ion insertion[38]. Despite the much lower specific capacity, NVO displays an obviously larger shift (0.30°, interlayer expansion by 2.4 Å, Fig. 4c) than NVO/2DPM (0.24°, interlayer expansion by 1.9 Å, Fig. 4d) during the discharge process. More interestingly, the (204) peak of NVO quickly reaches its lowest value and remains stable over a wide potential range. By contrast, the (204) peak of NVO/2DPM gradually shifts towards negative value and reaches a minimum value at 0.3 V vs. Zn/Zn$^{2+}$. Furthermore, V K-edge X-ray absorption spectra of NVO and NVO/2DPM at the fully charged and discharged states were also collected (Supplementary Figs. 33 and 34). The local coordination

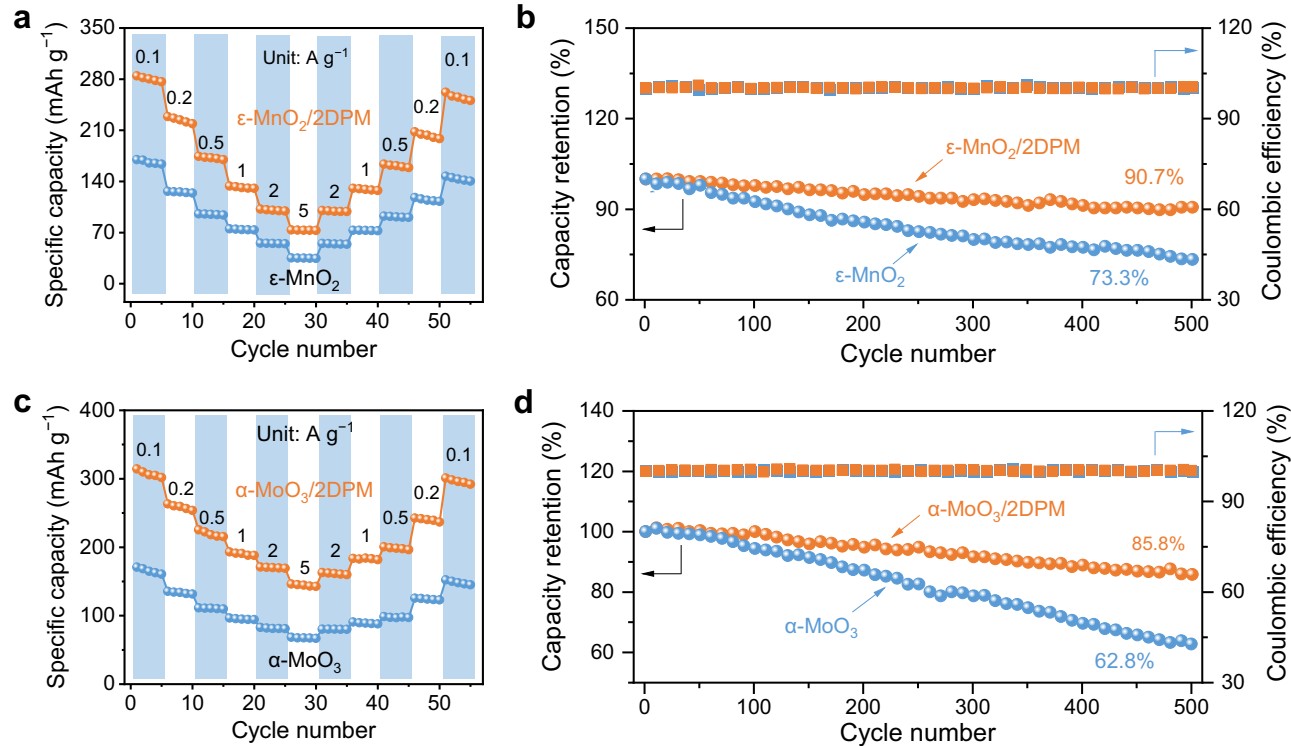

**Fig. 5 | Effects of 2DPM on different cathodes. a** Rate performance and **b** cycling performance at 3 A g⁻¹ of ε-MnO₂ and ε-MnO₂/2DPM in 2 M ZnSO₄. **c** Rate performance and **d** cycling performance at 3 A g⁻¹ of α-MoO₃ and α-MoO₃/2DPM in 20 m ZnCl₂.

environment of V in both NVO (Fig. 4e) and 2DPM/NVO (Fig. 4f) is disclosed by the wavelet-transform analysis of the extended X-ray absorption fine structure spectra (EXAFS) in $R$ and $k$ spaces. In NVO, the intensity maximum corresponding to the first-shell V−O scattering shows a considerable difference in the k space between the charged (8.8 Å⁻¹) and discharged (10.1 Å⁻¹) state, indicating a large increase in V−O bond length induced by ion insertion. In contrast, a much smaller change in k space can be identified for the intensity maximum of NVO/2DPM between the charged (8.8 Å⁻¹) and discharged (9.1 Å⁻¹) states. All these observations suggest that the increased H⁺ charge carriers assist in alleviating the repulsive interaction between the cathode host and charge carriers, thereby mitigating the volume change and structural distortion of the electrode.

### Universal effect of the 2DPM coating on other H⁺/Zn²⁺ co-insertion cathodes

The success of the ion-filtering 2DPM coating strategy in enhancing the charge-storage kinetics and durability of NVO motivated us to assess its universality for different AZB cathodes. We thereby synthesized two additional recognized cathode materials, namely tunnel-type ε-MnO₂ (Supplementary Fig. 35)[16] and layer-structure α-MoO₃ (Supplementary Fig. 36)[39]. They were subsequently prepared into high-mass-loading electrodes and evaluated in the dilute ZnSO₄ electrolyte (2 M) and the highly concentrated ZnCl₂ (20 m) electrolyte, respectively. We compared the electrochemical performance of ε-MnO₂ and α-MoO₃ with their corresponding 2DPM-80-covered electrodes (denoted ε-MnO₂/2DPM and α-MoO₃/2DPM). As shown in Supplementary Fig. 37, ε-MnO₂/2DPM shows similar shapes in their CV and GCD curves to pristine ε-MnO₂. The specific capacity of ε-MnO₂/2DPM (5 mg cm⁻²) reaches 280.3 mAh g⁻¹ at 0.1 A g⁻¹ (theoretical capacity of 307.9 mAh g⁻¹), which represents a capacity enhancement of 69.6% compared with pristine ε-MnO₂ (165.3 mAh g⁻¹, Fig. 5a). After 500 charge/discharge cycles at 3 A g⁻¹, ε-MnO₂/2DPM sustained 90.7% of its original capacity, while the capacity retention of ε-MnO₂ only reached

73.3% (Fig. 5b). Likely, 2DPM is effective to improve the specific capacity and cycling performance of α-MoO₃ (Supplementary Fig. 38). α-MoO₃/2DPM (10 mg cm⁻²) displays a specific capacity of 310 mAh g⁻¹ at 0.1 A g⁻¹ (theoretical capacity of 372.1 mAh g⁻¹), which significantly contrasts with the low specific capacity of pristine α-MoO₃ (164.9 mAh g⁻¹, Fig. 5c). Besides, with the incorporation of 2DPM, the capacity retention of α-MoO₃ increases from 62.8% to 85.8% after 500 charge/discharge cycles at 3 A g⁻¹.

## Discussion

In summary, we have showcased a proton-selective interfacial coating strategy for achieving high-mass-loading AZB cathodes by employing 2DPM with dual ion-transport nanochannels and dense proton-conduction groups. The 2DPM coating substantially promoted H⁺ passing at the cathode/electrolyte interface, thus enabling the cathode electrochemistry transition from Zn²⁺-dominated to H⁺-dominated Faradic reactions. As a result, we achieved high-mass-loading NVO/2DPM with considerably boosted specific capacity (288.8 vs. 450.5 mAh g⁻¹), high areal capacity (4.5 mAh cm⁻²), and enhanced cycling stability (68.6% vs. 87.8% after 1000 cycles) compared to NVO electrode. We further show that this 2DPM could be universally applicable to different cathodes (i.e., ε-MnO₂ and α-MoO₃) and aqueous electrolytes (i.e., 2 M ZnSO₄ and 20 m ZnCl₂) of AZBs. The value behind the exemplified ion-filtering coating for AZBs is apparent, as it helps maximise the expression of the theoretical charge-storage promise of diverse cathodes with a practical mass loading. In addition, the fundamental insights gained from interfacial ion regulation will provide essential guidelines for designing sustainable and high-performance aqueous batteries relying on diverse charge carrier ions. We should note that the H⁺-involved cathode reaction could lead to changes in the electrolyte environment, accelerating the parasitic reactions of the Zn metal anode. Particular attention should also be paid to protecting Zn metal anodes when our reported coating is employed for full device assembly. To this end, a range of previously

reported strategies could be adopted, such as interphase construction and electrolyte additives with pH-adaptive capability. The interfacial coating of 2D polymer membrane could also be a promising strategy to address the challenges of Zn metal anodes. In this sense, high $Zn^{2+}$ conductivity/selectivity and hydrophobicity will be pursued criteria for the 2D polymer coating. Rationally designing and synthesizing new 2D polymer membranes for Zn metal anodes remains an interesting direction for future exploration.

## Methods

### Materials

5, 10, 15, 20-tetrakis (4-aminophenyl) porphyrin (TAPP), 2,5-dihydroxyterephthalaldehyde (DHTAP), and 5, 10, 15, 20-tetrakis (4-aminophenyl) porphyrin-Cu (II) were obtained from TCI Deutschland GmbH (Germany), Porphyrin Laboratories GmbH (Germany), and Sigma-Aldrich, respectively. Sodium oleyl sulfate (SOS), vanadium pentoxide ($V_2O_5$), NaCl, $ZnSO_4$, and Zn foils were all purchased from Sigma-Aldrich. Carbon black, polyvinylidene fluoride (PVDF), trifluoromethanesulfonic acid, 1-methyl-2-pyrrolidone (NMP), and chloroform were provided by Alfa Aesar. Carbon paper was purchased from Fuel Cell Store. All chemicals were used directly without any purification.

### Synthesis of 2DPM

2DPM was synthesized according to our previously reported surfactant-monolayer-assisted interfacial synthesis method[34]. Briefly, 50 mL of ultrapure Milli-Q water was added into a crystallising dish with a diameter of 50 mm. Then, 20 μL of SOS surfactant (1 mg mL$^{-1}$ in chloroform) was gently spread onto the water surface using a microsyringe. After evaporating the chloroform solvent (1 h), the surfactant could completely cover the water surface. Next, 0.5 mL of TAPP solution (1 mg mL$^{-1}$ in trifluoromethanesulfonic acid) was slowly injected into the water phase, followed by waiting for 1 h to fully disperse TAPP in the water. After that, 3.4 mL solution of DHTAP (1 mg mL$^{-1}$ in water) was injected into the water phase. Finally, the reaction was kept at 50 °C for 5 days. The thickness could be regulated by adding stoichiometric monomers.

### Preparation of NVO nanowires

1 g of $V_2O_5$ powder was added into 15 mL of the NaCl aqueous solution with a concentration of 2 mol L$^{-1}$. Then, the mixture was stirred for 96 h at 30 °C to finish the reaction. After washing with deionized water for several times and freeze-drying, the black red product was obtained[38].

### Preparation of the ε-MnO$_2$ electrode

The ε-MnO$_2$ electrode was synthesized by an electrodeposition method according the previous reported work[16]. A three-electrode cell for electrodeposition was specially fabricated, comprising of a working electrode (carbon paper), counter and reference electrodes (Zn foil), and aqueous electrolyte (2 m $ZnSO_4$, 0.2 m $MnSO_4$). Firstly, the three-electrode cell was galvanostatically charged at 0.2 mA cm$^{-2}$ to 1.8 V (vs Zn/Zn$^{2+}$) using a CHI 660E electrochemical workstation. After maintained 1.8 V for 24 h, ε-MnO$_2$ could electrodeposit on the surface of carbon paper with a mass loading of about 5 ± 1 mg cm$^{-2}$. The electrodeposition weight could be controlled by the electrodeposit time. Then, the resultant MnO$_2$@carbon paper was dried in a vacuum oven at 80 °C overnight and directly cut into many discs as the electrodes.

### Preparation of α-MoO$_3$ nanowires

α-MoO$_3$ nanowires were prepared by a hydrothermal method according to a previous report[39]. Briefly, 6.5 g of $(NH_4)_6Mo_7O_{24}\cdot 4H_2O$ was first dissolved into 180 mL of deionized water, and 30 mL of concentrated HNO$_3$ (70 wt%) was subsequently added with strong stirring. Then, the mixed solution was transferred into a 250 mL Teflon-lined stainless autoclave and heated in an electric oven at 180 °C for 12 h. After cooled down to 25 °C, the resultant powder sample was washed with distilled water and dried in a vacuum oven at 80 °C for 12 h.

### Characterisation

The chemical structures of 2DPM were analysed by FT-IR Spectrometer Tensor II (Bruker) with a universal Zn-Se ATR (attenuated total reflection). SEM images were captured using Gemini 500 (Zeiss, Germany) and AFM images were performed on Multimode-8 (Bruker, USA). Optical microscope (Axioscope 5, Zeiss, Germany) was conducted to study the morphology of 2DPM. HR-TEM and SAED characterisations were carried out using Carl Zeiss Libra 200 Cs operated at the acceleration voltage of 200 KV. High-angle annular dark-field scanning transmission electron microscopy (HAADF-STEM) and EDX measurements were performed using a detector of Oxford Instrument attached to the TEM. ICP-AES (Avio 220 Max, Perkin Elmer corporation, USA) was used to test the ion concentration of Zn$^{2+}$ and V-species. XAS measurements were conducted to record the K edge of V (5460-5500 eV) at beamline P65 (DESY, Hamburg, Germany). The XAS results were processed and analysed by using Demeter software[40]. GIWAXS measurements were performed at the 3 C SAXS-I and 9A U-SAXS beamline at PLS-II (Pohang Accelerator Laboratory), Pohang, Republic of Korea. The detector was a 2D CCD Detector Rayonix SX165, USA and the beam energy was 11.08 keV (λ = 1.12 Å). The sample-detector distance was verified to be 221.81 mm by using silver behenate (AgBH) as a calibration standard. The incidence angle of the beam was chosen to be 0.10° and the exposure time to the beam was 10 s with 2.5 degree of attenuation. The resulting images were then analysed with Igor GIWAXSshop code.

### Ion permeation measurements

A H-shaped device with two chambers connected by an open hole of 4 mm in the centre was customised to test the ion sieving performance of the nanomembranes (Supplementary Fig. 9). Nylon microporous membrane with a pore size of 0.45 μm was applied to support 2DPM with an area of 2 cm$^2$, and then mounted them between two filter chambers. One of the chambers was filled with 20 mL of high-concentration salt solution (0.05 mol L$^{-1}$ H$_2$SO$_4$ or 0.1 mol L$^{-1}$ ZnSO$_4$), setting as the seed solution. The other one was permeate chamber filling with 20 mL of deionized water. In order to avoid concentration-induced polarization, magnetic stirring was conducted in both chambers. The H$^+$ permeation curves of 2DPM can be straightforwardly derived by recording the pH change in the permeated chamber at regular intervals over time. Besides, Zn$^{2+}$ transport, as the competition with H$^+$ transport in AZBs, was also assessed for 2DPM by substituting 0.05 M H$_2$SO$_4$ with 0.1 M ZnSO$_4$ in the seed chamber. The Zn$^{2+}$ concentration in the permeated chamber was monitored by ICP-AES. The ion permeation rate ($J$, mol m$^{-2}$ h$^{-1}$) could be calculated using Eq. (1)[41], where $C$ (mol L$^{-1}$) and $V$ (L) are the concentration and volume of the solution in the permeate chamber, respectively. $S$ is the membrane area (m$^2$), and $\Delta t$ (h) is the test time. The H$^+$/Zn$^{2+}$ selectivity of 2DPM was calculated by the ratio of H$^+$ and Zn$^{2+}$ permeation rate. Diffusion coefficients of permeated ions ($D$, m$^2$ h$^{-1}$) were calculated based on the classical diffusion Eq. (2)[42], where $d$ (m) is the thickness of 2DPM, $A$ (m$^2$) is the pore area of 2DPM. Moreover, the ion conductivity of 2DPM-80 was tested by using Ag/AgCl electrodes placed inside two chambers. Both chambers were filled with 0.5 M H$_2$SO$_4$. The voltage range is from −0.2 to 0.2 V with a step size of 0.01 V s$^{-1}$. Then, the current was recorded as a function of the applied voltage by a CHI 660E electrochemical workstation.

$$J = \frac{C \cdot V}{S \cdot \Delta t} \qquad (1)$$

$$D = \frac{J \cdot d \cdot S}{C \cdot A} \qquad (2)$$

## Electrochemical measurements

To prepare 2DPM-covered cathodes, active material (NVO or $\alpha$-MoO$_3$), binder (PVDF), and carbon black were firstly mixed with a weight ratio of 7:2:1 in NMP. The mixed slurry was coated on a carbon paper, followed by drying at 80 °C for 12 h. The mass loading of NVO and $\alpha$-MoO$_3$ was measured to be about $10 \pm 1$ mg cm$^{-2}$. Then, 2DPM was transferred on the electrode surface by a liquid-transfer method. The resultant electrodes were cut into many discs with a diameter of 8 mm. The electrochemical cells were assembled by with a two-electrode Swagelok cell using the prepared disk electrodes as cathode (NVO, $\varepsilon$-MnO$_2$, or $\alpha$-MoO$_3$), Zn foil as the anode, glass-fibre filter as the separator, 2 M ZnSO$_4$ or 20 m ZnCl$_2$ aqueous solution as the electrolyte. Zn metal foil was polished by a sandpaper and cut into a disk anode with a diameter of 6 mm. The stainless steel and titanium rods were adopted as the current collectors of anode and cathode, respectively. AZB pouch cells were assembled by using two NVO/2DPM cathodes with Al current collector (2.5 × 4 cm$^2$), Zn foil anode (2.5 × 4 cm$^2$, 10 µm, 0.1 g), and glass fibre separator (200 µm). Such a two-cathode configuration is primarily employed to minimise the capacity gap between the anode and cathode in the device, aiming to achieve a suitable anode/cathode mass ratio. First, active materials were deposited on the Al foil with an areal density of $10 \pm 1$ mg cm$^{-2}$ for the cathode and covered by 2DPM coating. Subsequently, the electrodes were compacted and then dried at 65 °C under vacuum for 72 h, followed by laminating, welding, wrapping, baking, injecting electrolyte, and pre-packaging. 2 mL electrolyte were added. Before the cycling test, the assembled pouch cell was firstly cycled for 5 times at 0.5 A g$^{-1}$ to fully wet the thick electrodes and release the generated gas via opening the packing film. After that, the cell was re-sealed and conducted the cycling tests. Both CV and EIS measurements were performed on a VMP3 potentiostat (Biologic, France). The EIS measurements were carried out at a 20 mV AC oscillation amplitude over the frequency range of 100 kHz to 0.01 Hz. The GCD curves, cycling stability tests, and GITT measurements were conducted on a Land battery test system (LAND CT2001A) at 25 °C. For the GCD and CV tests, the potential range was set from 0.3 to 1.5 V vs. Zn/Zn$^{2+}$ and the current density varied from 0.1 to 5 A g$^{-1}$. For the cycling tests, the prepared cells were placed in the climatic chamber with a constant temperature of 25 °C. GITT measurements was performed with a galvanostatic charge/discharge pulse of 0.1 A g$^{-1}$ for 10 min, followed by an open circuit for 1 h. Energy density ($E$) was calculated using the GCD profile based on Eq. (3), where $I$ is the applied current density based on the mass of cathode materials, $U$ is the cell output voltage and $t$ is the discharging time.

$$E = I \int_0^t U(t)dt \tag{3}$$

## EQCM measurements

The NVO slurry was coated on the Au-coated quartz crystal and dried at 80 °C for 12 h. Owing to the range limitation, the EQCM can only test the electrode with a small mass loading (0.1 mg cm$^{-2}$). The EQCM cell was assembled using Zn foil as the anode and 2 M ZnSO$_4$ as the electrolyte. Next, the EQCM tests were performed on the Gamry EQCM 10 M™ devices (Gamry Instruments, USA). CV was conducted at a scan rate of 1 mV s$^{-1}$ at the potential range between 0.3 and 1.5 V vs. Zn/Zn$^{2+}$. The frequency change as a function of potential was recorded in real time.

## Calculation for H$^+$/Zn$^{2+}$ insertion ratio

After 3 charge/discharge cycles at 0.1 A g$^{-1}$, various fully discharged NVO/2DPM electrodes were washed with deionized water and then sonicated for 3 h in 2 wt% HNO$_3$ solution to obtain transparent discharged NVO solution. The total electron transfer number ($n$) per

stoichiometric unit of NaV$_3$O$_8$·1.5H$_2$O can be obtained using Eq. (4), where $C_1$ is the measured specific capacity, and $C_2$ is the theoretical specific capacity of NaV$_3$O$_8$·1.5H$_2$O. Meanwhile, the Zn/V atomic ratio ($R_{Zn/V}$) in the fully discharged electrode is quantified by the ICP-AES analysis. With $n$ and $R_{Zn/V}$, the H/V atomic ratio ($R_{H/V}$) in the fully discharged electrode can be estimated according to Eq. (5). Finally, the H$^+$/Zn$^{2+}$ insertion ratio ($R_{H/Zn}$) can be derived based on Eq. (6)[38].

$$n = \frac{6C_1}{C_2} \tag{4}$$

$$R_{H/V} = \frac{n - 6R_{Zn/V}}{3} \tag{5}$$

$$R_{H/Zn} = \frac{R_{H/V}}{R_{Zn/V}} \tag{6}$$

## Operando synchrotron XRD measurements

2025-type coin cells with an open hole in the middle were specially prepared for the operando XRD tests, which allow the incidence of X-ray. A titanium foil with same hole was placed between coin cell case and cathode to avoid corrosion, and Kapton tape was used to seal the hole. The Biologic VMP3 potentiostat was utilized to control the operando charge/discharge process. Operando XRD measurements were carried out at beamline P02.1 from DESY synchrotron source (PETRA III, Hamburg, Germany) with a wavelength of 0.20733 Å. The diffraction data was recorded by a Perkin Elmer 2D detector in real time.

## DFT calculations

The first-principles calculations were carried out with the Vienna ab initio simulation package (VASP 5.4.4)[43,44]. The interaction between ions and valence electrons was described using projector augmented wave (PAW) potentials, and the exchange-correlation between electrons was treated through using the generalised gradient approximation (GGA) in the Perdew-Burke-Ernzerhof (PBE) form[45]. DFT-D3 method was employed to calculate the van der Waals (vdW) interaction[46]. The plane wave cutoff energy was 450 eV, and a k-point grid of 1 × 1 × 7 and 1 × 1 × 1 was used for bulk and slab models. Ionic relaxations were carried out under the conventional energy (10$^{-4}$ eV) and force (0.03 eV Å$^{-1}$) convergence criteria. The climbing image nudged elastic band (CI-NEB) method was used to locate the diffusion pathway[47,48]. Activation barriers were calculated by the energy differences between the transition and initial states. The theoretical approach was based on the GGA method with on-site Coulomb interaction parameter (GGA + U method), in which an effective U-J parameter of 6.3 eV was applied to improve the description Zn 3d states[49]. Note that the water environment was treated implicitly using the VASPsol code with a dielectric constant of 78.4[50].

## Data availability

The data that support the findings of this study are available from the corresponding authors upon reasonable request. Source data are provided with this paper.

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

## Acknowledgements

This work was financially supported by European Union's Horizon Europe research and innovation programme (ERC Starting Grant, BattSkin, 101116722, M.Y.), European Union's Horizon 2020 research and innovation programme (LIGHT-CAP 101017821, X.F.), the ERC Consolidator Grant (T2DCP, 819698, X.F.), German Research Foundation (DFG) within the Cluster of Excellence, CRC 1415 (Grant No. 417590517, X.F.), and Polymer-based Batteries (SPP 2248, RACOF-MMIS, X.F.). Q.G. was supported by a grant from the China Scholarship Council (File No. 202006240244). The authors acknowledge the use of the facilities in the Dresden Center for Nanoanalysis (DCN) at the Technische Universität Dresden, the GWK support for providing computing time through the Center for Information Services and High-Performance Computing (ZIH) at TU Dresden, beam time allocation at beamline P02.1 and P65 at the PETRA III synchrotron (DESY, Hamburg, Germany), and beam time allocation at beamline 3C SAXS-I and 9A U-SAXS at the Pohang Accelerator Laboratory (PLS-II). The authors also thank Dr. Panpan Zhang (TU Dresden), Dr. Gang Wang (TU Dresden), Dr. Mingchao Wang (TU Dresden), and Xiaohui Liu (TU Dresden) for helpful discussions.

## Author contributions

Q.G., M.Y., and X.F. conceived the project and designed materials and experiments. Q.G. conducted material preparation, battery assemble, electrochemical testing, and most characterisations. W.L. synthesized 2DPM. X.L. performed the DFT calculations. J.X.Z., D.S., J.J.Z., D.L., J.D., and X.C. conducted important characterisations and prepared the schematics. B.Z. and Z.L. conducted the HR-TEM, TEM mapping, and SAED investigation. N.N., S.C., and K.C. performed the GIWAXS measurement. Z.Z. supervised the concentration-driven ion permeation tests. X.Z., G.S., and T.H. contributed to the important discussion. Q.G. and M.Y. wrote the whole manuscript. All authors contributed to revise the manuscript under the supervision of M.Y. and X.F.

## Funding

## Competing interests

The authors declare no competing interests.
