## [Peer Review File · Nature Communications]

Proton-Selective Coating Enables Fast-kinetics High-Mass-Loading Cathodes for Sustainable Zinc BatteriesREVIEWER COMMENTS

Reviewer #1 (Remarks to the Author):

In this study, the researchers have detailed the development of an ultra-thin 2D polyimine nanomembrane (2DPM) featuring innovative dual ion-transport nanochannels and densely distributed proton-conduction groups. This configuration facilitates a heightened ion flux and exceptional H⁺/Zn²⁺ selectivity at the cathode/electrolyte interface within aqueous zinc batteries. The thickness of the 2DPM allows for precise regulation of the H⁺ and Zn²⁺ charge carrier ratio, effectively addressing the challenge of Zn²⁺ insertion in the cathode and realizing theoretical-close electrochemical performance. The introduction of the 2DPM presents an intriguing and profound strategy for achieving controllable charge carrier regulation in zinc batteries, leveraging the atomic-precise nature of the membrane to greatly enhance charge-storage capacity and durability. This research holds significant scientific merit in the context of battery studies, offering valuable insights for the design of high-performance batteries with a universal artificial cathode/electrolyte interface. However, for a more comprehensive understanding and to address potential concerns, I would recommend some minor revisions before acceptance by Nature Communications. Additionally, several questions have been raised:

1. Beyond addressing the cathode challenges in aqueous zinc batteries, it would be insightful to explore the potential application of this strategy to improve the stability of the zinc anode, particularly concerning issues such as stripping/plating and dendrite formation. Is it possible to extend this strategy for Zinc anode?
2. Given the ultrathin nature of the 2DPM and its linkage through imine bonds, concerns about stability during repeated electrochemical processes have been raised. It is essential to include experimental data assessing the electrochemical stability of the individual 2DPM to provide a comprehensive evaluation of its performance over time.
3. Clarification is sought regarding the choice of a two NVO/2DPM cathode configuration in the pouch cell demonstration, as illustrated in Supplementary Fig. 27. Could you please give some explanation about this specific design?
4. To ensure the work is up-to-date and relevant in the context of recent advancements in the field, it is recommended to cite recent important publications, such as Nat. Commun. 2022, 13, 5348; Nat. Commun. 2023, 14, 6526 (2023); Angew. Chem. 2023, 135, e202307.
5. To address confusion regarding the calculation of the H⁺/Zn²⁺ insertion ratio, it would be better to provide a thorough explanation of the calculation methodology.

Reviewer #2 (Remarks to the Author):

Guo et al. present an innovative approach involving a 2D polyimine nanomembrane-based proton-selective coating of Zn battery cathodes to achieve fast kinetics and high mass loading. The artificial 2DPM coating with well-designed structures demonstrates a notable H⁺ flux and an excellent H⁺/Zn²⁺ transport selectivity, transitioning the cathode mechanism from sluggish Zn²⁺-dominated to fast-kinetics H⁺-dominated intercalation electrochemistry. Consequently, standard NVO cathode combined with 2DPM exhibits a theoretically close capacity, exceptionally high areal capacity, state-of-the-art energy density, and improved cycling stability. Also, the universal effect of such a coating strategy is also validated by assessing two other cathode materials. Overall, the proton-selective coating strategy for Zn battery cathodes is new and appears highly effective. Most statements and conclusions of

this work are convincing with solid experimental support. Therefore, I find this study novel and impactful enough for the publication in Nat. Commun. However, my below concerns should be addressed before acceptance.

1. In addition to achieving high ion transport flux and excellent ion selectivity, an ideal interfacial coating should also possess poor electron conductivity. The authors should provide information and related discussion on the electron conductivity of 2DPM?
2. Structural stability of 2DPM is surely a critical consideration during repeated charge and discharge processes. The authors should present the electrochemical stability result of 2DPM after cycling?
3. For practical applications, a high mass loading of active material is necessary, ideally exceeding 10 mg cm^{-2} . Did the authors test the specific capacity variation of the $\text{NaV}_3\text{O}_8 \cdot 1.5\text{H}_2\text{O}$ cathode at different mass loadings? It is appreciated to show the relevant result.
4. In DFT calculations, it is crucial to consider water molecules, as ions transport in a water environment. Did the authors simulate the water conditions in the DFT calculations? The related discussion should be added.
5. Minor issue: the authors are suggested to keep the consistency in abbreviations throughout the manuscript, such as NVO with 80 nm 2DPM. There is no need to re-denote 2DPM-80-covered NVO in the boosted charge-storage kinetics section.

Reviewer #3 (Remarks to the Author):

The authors describe in the manuscript an ultrathin 2D polyimine membrane (2DPM) having dual ion-transport nanosized channels and densely distributed proton conductive groups for aqueous zinc batteries. They demonstrate high proton flux and superior proton/zinc cation transport selectivity in permeation tests of the titled 2DPM. The unique features of the membrane are proposed to lead to fast kinetics of proton-dominated Faradic reactions within high mass loading of sodium vanadium oxide cathodes. Besides, high areal capacity and energy density as well as remarkable cycling stability are reported. Furthermore, the concept of interfacial proton-selective transport is demonstrated to be applicable to other cathodes and aqueous electrolytes. Overall, this is a piece of nice work reporting a novel membrane coating to regulate ionic transport for aqueous multivalent metal ion batteries. I'd like to recommend publication of this study after further considering the following points.

- 1, It should be noted that the concentration of proton in a 2M ZnSO_4 aqueous solution is very limited as compared to Zn cations. In this case, can proton act as a suitable charge carrier? When proton permeation is much faster than Zn cations, the concentration of proton will be in an unbalanced state, and thus cause drastic change of the local environment of the counter electrode (i.e., zinc anode).
- 2, What is the pH value and pH value change in the electrolyte during the discharge and charge of the cathode?
- 3, What about the effect of the membrane thickness on the ionic diffusivity/selectivity and thus the electrochemical performance?
- 4, The coating of ionic screening layer on cathode may impede electronic conduction. Any investigation along this line?
- 5, Is the superior electrode kinetics partially ascribed to the 1D features of the vanadate cathode materials?

To Reviewer 1:

In this study, the researchers have detailed the development of an ultra-thin 2D polyimine nanomembrane (2DPM) featuring innovative dual ion-transport nanochannels and densely distributed proton-conduction groups. This configuration facilitates a heightened ion flux and exceptional H^+/Zn^{2+} selectivity at the cathode/electrolyte interface within aqueous zinc batteries. The thickness of the 2DPM allows for precise regulation of the H^+ and Zn^{2+} charge carrier ratio, effectively addressing the challenge of Zn^{2+} insertion in the cathode and realizing theoretical-close electrochemical performance. The introduction of the 2DPM presents an intriguing and profound strategy for achieving controllable charge carrier regulation in zinc batteries, leveraging the atomic-precise nature of the membrane to greatly enhance charge-storage capacity and durability. This research holds significant scientific merit in the context of battery studies, offering valuable insights for the design of high-performance batteries with a universal artificial cathode/electrolyte interface. However, for a more comprehensive understanding and to address potential concerns, I would recommend some minor revisions before acceptance by Nature Communications. Additionally, several questions have been raised:

Response: We appreciate the positive comment of the reviewer. Additional experiments and revisions have been conducted according to your following comments.

1. Beyond addressing the cathode challenges in aqueous zinc batteries, it would be insightful to explore the potential application of this strategy to improve the stability of the zinc anode, particularly concerning issues such as stripping/plating and dendrite formation. Is it possible to extend this strategy for zinc anode?

Response: Thank you for the insightful comment. In principle, the interfacial coating of 2D polymer membrane could be a promising strategy to address the challenges of Zn metal anodes. Ideally, such a coating would function akin to a solid-electrolyte interphase, isolating the Zn metal from direct contact with the electrolyte. This preventive action would mitigate parasitic Zn corrosion and inhibit the hydrogen evolution reaction. Meanwhile, the coating should selectively transport Zn^{2+} , promoting a rapid and homogenous Zn^{2+} flux on the Zn metal surface. Such a strategy would guide uniform and non-dendritic Zn deposition. In this sense, high Zn^{2+} conductivity/selectivity and hydrophobicity will be pursued criteria for the 2D polymer coating¹⁻³, and the reported 2DPM in our study may not fully meet these criteria. Indeed, rationally designing and synthesizing 2D polymer membranes for Zn metal anodes remains an avenue for future exploration. We have added the corresponding discussion to the revised manuscript.

2. Given the ultrathin nature of the 2DPM and its linkage through imine bonds, concerns about stability during repeated electrochemical processes have been raised. It is essential to include experimental data assessing the electrochemical stability of the individual 2DPM to provide a comprehensive evaluation of its performance over time.

Response: In fact, the excellent electrochemical stability of 2DPM in mild acid electrolyte was well demonstrated in our study. Specifically, NVO/2DPM after the cycling test was disassembled from the cell and subjected to scanning electron microscopy (SEM) and Fourier transform infrared spectroscopy (FTIR) characterizations (Supplementary Fig. 29). As revealed, 2DPM remains tightly covering the NVO surface with all characteristic FTIR peaks detected, verifying the robust electrochemical stability of 2DPM during repeated charge/discharge cycles. To further address your concern, we have conducted a galvanostatic charge/discharge (GCD) measurement of a 2DPM-80 electrode (2DPM-80-covered carbon paper) in a 2-electrode Swagelok cell, employing Zn foil as the counter electrode and 2 M ZnSO₄ as the electrolyte. As depicted in **Figure R1**, 2DPM exhibits nearly identical GCD profiles, further verifying its excellent electrochemical stability within the potential range of 0.3~1.5 V vs. Zn/Zn²⁺. The corresponding discussion has been added into the revised manuscript.

Figure R1. The 1st-cycle, 50th-cycle, and 100th-cycle GCD curves of 2DPM-80 at 0.1 mA cm⁻².

3. Clarification is sought regarding the choice of a two NVO/2DPM cathode configuration in the pouch cell demonstration, as illustrated in Supplementary Fig. 27. Could you please give some explanation about this specific design?

Response: We appreciate the valuable question. The pouch cell demonstration serves as a proof of concept to illustrate the practicality of our interfacial coating strategy for large-scale battery devices. The chosen device configuration, utilizing two NVO/2DPM cathodes, was primarily employed to

minimize the capacity gap between the anode and cathode in the device, aiming to achieve a suitable N/P ratio. We have clarified it in our revised manuscript.

4. To ensure the work is up-to-date and relevant in the context of recent advancements in the field, it is recommended to cite recent important publications, such as Nat. Commun. 2022, 13, 5348; Nat. Commun. 2023, 14, 6526; Angew. Chem. 2023, 135, e202307.

Response: Thank you for sharing these pioneering studies. We have included these references in the revised manuscript.

5. To address confusion regarding the calculation of the H⁺/Zn²⁺ insertion ratio, it would be better to provide a thorough explanation of the calculation methodology.

Response: We appreciate the valuable suggestion of the reviewer. The H⁺/Zn²⁺ insertion ratio was determined following a method reported previously.⁴ Specifically, the total electron transfer number (n) per stoichiometric unit of NaV₃O₈·1.5H₂O can be obtained using **equation (R1)**, where C_1 is the measured specific capacity, and C_2 is the theoretical specific capacity of NaV₃O₈·1.5H₂O. Meanwhile, the Zn/V atomic ratio ($R_{Zn/V}$) in the fully discharged electrode is quantified by the inductively coupled plasma atomic emission spectroscopy (ICP-AES) analysis. With n and $R_{Zn/V}$, the H/V atomic ratio ($R_{H/V}$) in the fully discharged electrode can be estimated according to **equation (R2)**. Finally, the H⁺/Zn²⁺ insertion ratio ($R_{H/Zn}$) can be derived based on **equation (R3)**. The detailed explanation has been added into the revised manuscript.

$$n = \frac{6C_1}{C_2} \quad (\mathbf{R1})$$

$$R_{H/V} = \frac{n - 6R_{Zn/V}}{3} \quad (\mathbf{R2})$$

$$R_{H/Zn} = \frac{R_{H/V}}{R_{Zn/V}} \quad (\mathbf{R3})$$

To Reviewer 2

Guo et al. present an innovative approach involving a 2D polyimine nanomembrane-based proton-selective coating of Zn battery cathodes to achieve fast kinetics and high mass loading. The artificial 2DPM coating with well-designed structures demonstrates a notable H^+ flux and an excellent H^+/Zn^{2+} transport selectivity, transitioning the cathode mechanism from sluggish Zn^{2+} -dominated to fast-kinetics H^+ -dominated intercalation electrochemistry. Consequently, standard NVO cathode combined with 2DPM exhibits a theoretically close capacity, exceptionally high areal capacity, state-of-the-art energy density, and improved cycling stability. Also, the universal effect of such a coating strategy is also validated by assessing two other cathode materials. Overall, the proton-selective coating strategy for Zn battery cathodes is new and appears highly effective. Most statements and conclusions of this work are convincing with solid experimental support. Therefore, I find this study novel and impactful enough for the publication in Nat. Commun. However, my below concerns should be addressed before acceptance.

Response: We appreciate the positive comment of the reviewer. Additional experiments and discussions have been conducted to address the following concerns.

1. In addition to achieving high ion transport flux and excellent ion selectivity, an ideal interfacial coating should also possess poor electron conductivity. The authors should provide information and related discussion on the electron conductivity of 2DPM?

Figure R2. The simulated electronic band structure of 2DPM.

Response: Thank you for the constructive advice. The poor electron-conductive nature of 2DPM is evidenced by the simulated electronic band structure. As illustrated in **Figure R2**, a giant band gap of 1.53 eV is identified for 2DPM. Besides, the electron conductivity of 2DPM was experimentally assessed through a four-point probe measurement. The sheet resistance of 2DPM-80 reaches up to $6.1 \times 10^7 \text{ Ohm sq}^{-1}$, further underscoring its poor conductivity. The corresponding discussion has been included in the revised manuscript.

2. Structural stability of 2DPM is surely a critical consideration during repeated charge and discharge processes. The authors should present the electrochemical stability result of 2DPM after cycling?

Response: We appreciate the constructive suggestion of the reviewer. In fact, the excellent electrochemical stability of 2DPM was well demonstrated in our study. Specifically, NVO/2DPM after the cycling test was disassembled from the cell and subjected to scanning electron microscopy (SEM) and Fourier transform infrared spectroscopy (FTIR) characterizations (Supplementary Fig. 29). As revealed, 2DPM remains tightly covering the NVO surface with all characteristic FTIR peaks detected, verifying the robust electrochemical stability of 2DPM during repeated charge/discharge cycles. To further address your concern, we have conducted a galvanostatic charge/discharge (GCD) measurement of a 2DPM-80 electrode (2DPM-80-covered carbon paper) in a 2-electrode Swagelok cell, employing Zn foil as the counter electrode and 2 M ZnSO₄ as the electrolyte. As depicted in **Figure R3**, 2DPM exhibits nearly identical GCD profiles, further verifying its excellent electrochemical stability within the potential range of 0.3~1.5 V vs. Zn/Zn²⁺. The corresponding discussion has also been added to the revised manuscript.

Figure R3. The 1st-cycle, 50th-cycle, and 100th-cycle GCD curves of 2DPM-80 at 0.1 mA cm⁻².

3. For practical applications, a high mass loading of active material is necessary, ideally exceeding 10 mg cm⁻². Did the authors test the specific capacity variation of the NaV₃O₈·1.5H₂O cathode at different mass loadings? It is appreciated to show the relevant result.

Response: We appreciate the insightful suggestion. Following your suggestion, NVO/2DPM electrodes were fabricated with different mass loadings (*i.e.*, 2 ± 0.5, 5 ± 1, 10 ± 2, 20 ± 2, and 30 ± 3 mg cm⁻²). **Figure R4a** compares their GCD curves at a current density of 0.1 A g⁻¹. Along with the mass loading

increase from 2 to 30 mg cm⁻², the specific capacity slightly drops from 475.4 to 421.2 mAh g⁻¹. Meanwhile, the areal capacity experiences a significant enhancement from 0.95 mAh cm⁻² to an ultrahigh value of 12.64 mAh cm⁻² (**Figure R4b**). This result further highlights the effective role of the 2DPM coating in boosting the charge-storage kinetics of high-mass-loading cathodes for aqueous Zn batteries. The corresponding discussion has been added to the revised manuscript.

Figure R4. **a** GCD curves of NVO/2DPM electrodes with different mass loadings. **b** Corresponding areal capacity variation as a function of mass loading.

4. In DFT calculations, it is crucial to consider water molecules, as ions transport in a water environment. Did the authors simulate the water conditions in the DFT calculations? The related discussion should be added.

Response: In fact, the effect of water environment was considered in our DFT calculations. Specifically, we introduced a dielectric constant parameter (78.4) to mimic the water environment. This approach aligns with established practices in similar simulation studies⁵⁻⁷. We have added this explanation to the revised manuscript.

5. Minor issue: the authors are suggested to keep the consistency in abbreviations throughout the manuscript, such as NVO with 80 nm 2DPM. There is no need to re-denote 2DPM-80-covered NVO in the boosted charge-storage kinetics section.

Response: Thank you for the kind reminder. We have thoroughly checked all abbreviations and ensured that their definitions appear only once in the revised manuscript.

To Reviewer 3

The authors describe in the manuscript an ultrathin 2D polyimine membrane (2DPM) having dual ion-transport nanosized channels and densely distributed proton conductive groups for aqueous zinc batteries. They demonstrate high proton flux and superior proton/zinc cation transport selectivity in permeation tests of the titled 2DPM. The unique features of the membrane are proposed to lead to fast kinetics of proton-dominated Faradic reactions within high mass loading of sodium vanadium oxide cathodes. Besides, high areal capacity and energy density as well as remarkable cycling stability are reported. Furthermore, the concept of interfacial proton-selective transport is demonstrated to be applicable to other cathodes and aqueous electrolytes. Overall, this is a piece of nice work reporting a novel membrane coating to regulate ionic transport for aqueous multivalent metal ion batteries. I'd like to recommend publication of this study after further considering the following points.

Response: We appreciate the positive comment of the reviewer. Additional experiments and discussions have been conducted to address the following concerns.

1. It should be noted that the concentration of proton in a 2 M ZnSO₄ aqueous solution is very limited as compared to Zn cations. In this case, can proton act as a suitable charge carrier? When proton permeation is much faster than Zn cations, the concentration of proton will be in an unbalanced state, and thus cause drastic change of the local environment of the counter electrode (i.e., zinc anode).
2. What is the pH value and pH value change in the electrolyte during the discharge and charge of the cathode?

Response: Thank you for the constructive comments. Since Q1 and Q2 address similar concerns, we are combining our responses to these two comments.

Indeed, the initial concentration of H⁺ in Zn salt electrolytes is limited (5×10^{-5} mol L⁻¹ for 2 M ZnSO₄, pH = 4.3). However, it is essential to note that the water solvent in the electrolyte can act as a proton reservoir. The consumption of H⁺ in the electrolyte would disrupt the equilibrium of the hydrolysis reaction of Zn²⁺ (**equation (R4)**), triggering the generation of more H⁺ charge carriers for the cathode. According to your question (Q2), we further evaluated the electrolyte pH evolution during a discharge/charge cycle in a 2-electrode Swagelok cell. Due to the limited amount of the used electrolyte, we were not able to test the pH with a standard pH meter. Instead, pH paper strips were employed to assess the pH variation (**Figure R5**). As expected, the insertion of H⁺ into the cathode causes a slight pH increase within a range of 4~6.

We understand the reviewer's concern that the H^+ -involved cathode reaction could lead to changes in the electrolyte environment, accelerating the parasitic reactions of the Zn metal anode. Specifically, the involvement of H^+ charge carriers was identified as a crucial reason for the formation of the passivation layer (i.e., $Zn_4SO_4(OH)_6 \cdot 4H_2O$) on the Zn anode^{4,8,9}. However, considering the substantial benefits of H^+ charge carriers brought for the cathode, such as fast reaction kinetics, high mass loading, and large areal capacity, we believe that H^+ can be considered suitable charge carriers for cathodes. We certainly agree with the reviewer that particular attention should also be paid to protecting Zn metal anodes when full devices are assembled for practical applications. To this end, a range of previously reported strategies could be adopted, such as interphase construction and electrolyte additives with pH-adaptive capability¹⁰⁻¹⁵. All these discussions have been included in the revised manuscript.

Figure R5. The pH evolution in the electrolyte during one discharge/charge cycle of the NVO/2DPM electrode.

3. What about the effect of the membrane thickness on the ionic diffusivity/selectivity and thus the electrochemical performance?

Response: We appreciate the insightful question of the reviewer. In fact, the effects of the membrane thickness on the ion transport properties of 2DPM and the electrochemical performance of the NVO electrode have been systematically investigated in our study. We specifically evaluated the transmembrane transport of H^+ - and Zn^{2+} for 2DPM with different thicknesses. The membranes with thicknesses of 20, 60, 80, and 100 nm are denoted 2DPM-20, 2DPM-60, 2DPM-80, and 2DPM-100, respectively. **Figure R6a** and **Figure R6b** plot the H^+ and Zn^{2+} permeation curves of 2DPM as a function of time, respectively. All the membranes follow the linear permeation relationship with constant transport rates for both H^+ and Zn^{2+} . It is notable that 2DPM in the thickness range of 20 ~ 80 nm depicts

almost thickness-independent H^+ transport, showing a high H^+ permeation rate of $0.91 \sim 0.95 \text{ mol m}^{-2} \text{ h}^{-1}$ and an excellent proton diffusion coefficient of $6.2 \times 10^{-7} \sim 6.5 \times 10^{-7} \text{ cm}^2 \text{ s}^{-1}$. When the membrane thickness reaches 100 nm, the H^+ permeation rate slightly decreases to $0.73 \text{ mol m}^{-2} \text{ h}^{-1}$. By contrast, Zn^{2+} transport through 2DPM heavily depends on the membrane thickness, and the Zn^{2+} permeation rates for 2DPM-20, 2DPM-60, 2DPM-80, and 2DPM-100 are 0.51, 0.27, 0.0065 and $0.0062 \text{ mol m}^{-2} \text{ h}^{-1}$, respectively. **Figure R6c** further summarizes the ion permeation rates of all the membranes and defines the $\text{H}^+/\text{Zn}^{2+}$ permeation rate ratio as the ion-transport selectivity. As revealed, 2DPM-80 exhibits the best $\text{H}^+/\text{Zn}^{2+}$ selectivity of 140.7, while maintaining a high H^+ permeation rate.

Figure R6. **a** H^+ and **b** Zn^{2+} permeation curves in the concentration-driven permeation measurements of 2DPM-20, 2DPM-60, 2DPM-80, and 2DPM-100. **c** Ion permeation rates and $\text{H}^+/\text{Zn}^{2+}$ selectivity of various 2DPM membranes.

To evaluate the impact of membrane thickness on the electrochemical performance of NVO, we conducted GCD measurements at various current densities for the NVO electrodes coated with the 2DPM membrane. The calculated specific capacities of all electrodes are summarized in **Figure R7a**. All the 2DPM membranes could boost the charge-storage capability of NVO, and the improvement degree of the specific capacity follows the trend of $2\text{DPM-20} < 2\text{DPM-60} < 2\text{DPM-100} < 2\text{DPM-80}$. This trend is consistent with the $\text{H}^+/\text{Zn}^{2+}$ selectivity trend illustrated in **Figure R6c**. Meanwhile, the quantity of H^+ charge carriers is estimated by considering the Zn/V atomic ratio and the total charge transfer per V atom. **Figure R7b** displays the charge carrier ratios ($\text{H}^+/\text{Zn}^{2+}$) of all the electrodes. The contribution of H^+ charge carriers to the total charge storage of different electrodes matches well with the $\text{H}^+/\text{Zn}^{2+}$ selectivity trend of the employed 2DPM. Specifically, 2DPM-80 empowers VNO with the largest $\text{H}^+/\text{Zn}^{2+}$ ratio of 3.5, which contrasts with the pristine VNO with a low $\text{H}^+/\text{Zn}^{2+}$ ratio of 0.4. The high $\text{H}^+/\text{Zn}^{2+}$ ratio as the charge carriers exactly equal to the $\text{H}^+/\text{Zn}^{2+}$ selectivity of 2DPM-80 in 2 M ZnSO_4 as measured in **Figure R6c**. All these discussions can be found in the revised manuscript.

Figure R7. **a** Specific capacities at various current densities and **b** the charge carrier ratios (H⁺/Zn²⁺) of NVO covered by 2DPM with different thicknesses.

4. The coating of ionic screening layer on cathode may impede electronic conduction. Any investigation along this line?

Figure R8. Nyquist plots of **a** NVO and **b** NVO/2DPM at different potentials. **c** The equivalent circuit model used for the EIS data fitting. **d** ESR and R_{ct} of NVO and NVO/2DPM at different potentials.

Response: Thank you for your valuable comment. To evaluate the effect of 2DPM on the internal resistance, the electrochemical impedance spectroscopy (EIS) measurement was performed for NVO and NVO/2DPM at different potentials (**Figure R8a-b**). By fitting the EIS results with an equivalent circuit shown in **Figure R8c**, the equivalent series resistance (*ESR*) and charge-transfer resistance (*R_{ct}*) are determined for both electrodes. Typically, *ESR* reflects the internal or Ohmic resistance of the whole electrochemical system. As revealed in **Figure R8d**, *ESR* of NVO/2DPM is slightly lower than NVO, indicating that the 2DPM coating could slightly alleviate the interfacial contact resistance, instead of impeding the electronic conduction. More importantly, NVO/2DPM presents significantly lower *R_{ct}* than NVO, manifesting the enhanced charge-transfer efficiency of NVO/2DPM associated with the enriched H⁺ as charge carriers. We have added the corresponding discussion in the revised manuscript.

5. Is the superior electrode kinetics partially ascribed to the 1D features of the vanadate cathode materials?

Response: Thank you for the insightful question. Certainly, the 1D nano-sized morphology of the electrode material could partially account for the superior electrode kinetics. According to **equation (R5)**, the ion diffusion time (τ) within the electrode is dependent on both ion diffusion length (L) and ion diffusion coefficient (D).¹⁶ In this sense, the 1D nanosized morphology could enable a considerably shortened L . Meanwhile, the enriched H⁺ charge carriers could effectively boost D . Both factors contribute to the shortened τ , thereby promoting the electrochemical reaction kinetics. The corresponding discussion has been added to the revised manuscript.

$$\tau = \frac{L^2}{D} \quad (\mathbf{R5})$$

References:

1. Yuan, L. et al. Regulation methods for the Zn/electrolyte interphase and the effectiveness evaluation in aqueous Zn-ion batteries. *Energy Environ. Sci.* **14**, 5669-5689 (2021).
2. Zhang, Q. et al. Interfacial design of dendrite-free zinc anodes for aqueous zinc-ion batteries. *Angew. Chem. Int. Ed.* **59**, 13180-13191 (2020).
3. Zhao, Z. et al. Horizontally arranged zinc platelet electrodeposits modulated by fluorinated covalent organic framework film for high-rate and durable aqueous zinc ion batteries. *Nat. Commun.* **12**, 6606 (2021).
4. Wan, F. et al. Aqueous rechargeable zinc/sodium vanadate batteries with enhanced performance from simultaneous insertion of dual carriers. *Nat. Commun.* **9**, 1656 (2018).
5. Jia, B. et al. Indium Cyanamide for Industrial-Grade CO₂ Electroreduction to Formic Acid. *J. Am. Chem. Soc.* **145**, 14101-14111 (2023).
6. Zhang, J. et al. Accelerating electrochemical CO₂ reduction to multi-carbon products via asymmetric intermediate binding at confined nanointerfaces. *Nat. Commun.* **14**, 1298 (2023).
7. Mathew, K. et al. Implicit solvation model for density-functional study of nanocrystal surfaces and reaction pathways. *J. Chem. Phys.* **140**, 084106 (2014).
8. Huang, J. et al. Polyaniline-intercalated manganese dioxide nanolayers as a high-performance cathode material for an aqueous zinc-ion battery. *Nat. Commun.* **9**, 2906 (2018).
9. Zhao, Q. et al. Boosting the energy density of aqueous batteries via facile grothuss proton transport. *Angew. Chem.* **133**, 4215-4220 (2021).
10. Luo, M. et al. Dynamic Regulation of the Interfacial pH for Highly Reversible Aqueous Zinc Ion Batteries. *Nano Lett.* **23**, 9491-9499 (2023).
11. Lin, C. et al. Adaptive Ionization-Induced Tunable Electric Double Layer for Practical Zn Metal Batteries over Wide pH and Temperature Ranges. *ACS Nano* **17**, 23181-23193 (2023).
12. Jin, S. et al. Stabilizing Interface pH by Mixing Electrolytes for High-Performance Aqueous Zn Metal Batteries. *Small* **18**, 2205462 (2022).
13. Yang, Q. et al. Stabilizing interface pH by N-modified graphdiyne for dendrite-free and high-rate aqueous Zn-ion batteries. *Angew. Chem.* **134**, 202112304 (2022).
14. Ouyang, K. et al. Synergistic Modulation of In-Situ Hybrid Interface Construction and pH Buffering Enabled Ultra-Stable Zinc Anode at High Current Density and Areal Capacity. *Angew. Chem.* **135**, e202311988 (2023).
15. Lyu, Y. et al. Organic pH Buffer for Dendrite-Free and Shuttle-Free Zn-I₂ Batteries. *Angew. Chem. Inter. Ed.* **62**, e202303011 (2023).
16. Xia, H. et al. A figure of merit for fast-charging Li-ion battery materials. *ACS Nano* **16**, 8525-8530 (2022).

REVIEWERS' COMMENTS

Reviewer #1 (Remarks to the Author):

I would like to support the publication of this manuscript

Reviewer #2 (Remarks to the Author):

The authors have faithfully revised the manuscript. It can be accepted with this version.

Reviewer #3 (Remarks to the Author):

The authors have nicely addressed referees' comments by supplementing more experimental data such as morphology evolution and pH probing to make the discussion more convincing. From my side, this updated version is publishable.